# Demonstration of Near Edge X-ray Absorption Fine Structure Spectroscopy of Transition Metals Using Xe/He Double Stream Gas Puff Target Soft X-ray Source

**DOI:** 10.3390/ma14237337

**Published:** 2021-11-30

**Authors:** Tomasz Fok, Przemysław Wachulak, Łukasz Węgrzyński, Andrzej Bartnik, Michał Nowak, Piotr Nyga, Jerzy Kostecki, Barbara Nasiłowska, Wojciech Skrzeczanowski, Rafał Pietruszka, Karol Janulewicz, Henryk Fiedorowicz

**Affiliations:** 1Institute of Optoelectronics, Military University of Technology, 2 Kaliskiego Str., 00-908 Warsaw, Poland; tomasz.fok@wat.edu.pl (T.F.); lukasz.wegrzynski@wat.edu.pl (Ł.W.); andrzej.bartnik@wat.edu.pl (A.B.); michal.nowak@wat.edu.pl (M.N.); piotr.nyga@wat.edu.pl (P.N.); jerzy.kostecki@wat.edu.pl (J.K.); barbara.nasilowska@wat.edu.pl (B.N.); wojciech.skrzeczanowski@wat.edu.pl (W.S.); karoljanulewicz51@gmail.com (K.J.); henryk.fiedorowicz@wat.edu.pl (H.F.); 2Institute of Physics, Polish Academy of Sciences, Aleja Lotnikow 32/46, 02-668 Warsaw, Poland; pietruszka@ifpan.edu.pl

**Keywords:** NEXAFS, X-ray absorption spectroscopy, soft X-rays, X-ray techniques, transition metals spectroscopy

## Abstract

A near 1-keV photons from the Xe/He plasma produced by the interaction of laser beam with a double stream gas puff target were employed for studies of L absorption edges of period 4 transitional metals with atomic number Z from 26 to 30. The dual-channel, compact NEXAFS system was employed for the acquisition of the absorption spectra. L_1–3_ absorption edges of the samples were identified in transmission mode using broadband emission from the Xe/He plasma to show the applicability of such source and measurement system to the NEXAFS studies of the transition metals, including magnetic materials.

## 1. Introduction

Material investigations are typically performed employing beamlines at synchrotrons. Such studies are often based on an investigation of subtle changes in the absorption coefficient spectrum in the X-ray and soft X-ray (SXR) spectral ranges near the absorption edge, i.e., using near edge X-ray absorption fine structure spectroscopy (NEXAFS) [1,2]. Measurement of changes in the absorption spectrum near the absorption edge of a given material allows for obtaining information not only about its elemental composition, but also electronic structure, oxidation states, as well as a local bonding environment [3]. So far, synchrotrons make it possible to test most of the materials near the K and L edges [4]. However, one of the biggest disadvantages of this type of measurement is the availability of the synchrotron measurement time, i.e., the need to apply for beamtime without a guarantee of obtaining such a possibility, and the measurement time itself is strictly defined, while the cost of such measurements for i.e., commercial applications, is very high.

Due to the above arguments, several research groups have started work and put significant effort into the development of the laboratory measurement systems for NEXAFS analysis. Such solutions are based on X-ray lamps [5,6] or laser-plasma EUV and SXR radiation sources [7,8,9,10].

Laser plasma sources, depending on the used combination of irradiation parameters (pulse energy, pulse duration, focusing parameters) and target type (solid or gas), generate enough radiation to conduct the NEXAFS measurements. However, such sources are typically operating at the photon energy range, around C-K edge (284 eV), up to O-K edge (~540 eV) [11] and are better suited towards biological (organic) applications, since those two edges are defining the “water window” spectral range [12]. Recently, compact laboratory systems emitting near 1-keV and above from laser-plasma sources were also demonstrated, enabling the study of heavier elements near their L absorption edges. These sources, however, are based on metal targets (i.e., copper Cu [7]) producing debris that cause problems due to grating and filter contamination or even damage and were equipped with spectrometers utilizing separate reference and sample spectra (channels) acquisition, for example, with gratings based on off-axis zone-plates [13]. The separate channels may cause errors while obtaining the optical density spectrum because a slight uncontrolled spectral shift will produce artifacts in the final absorption spectrum. Nevertheless, they can be successfully used to study heavier elements.

Among these heavier elements, there are several transition metals of particular interest to researchers, i.e., Fe, Co, Ni, Cu, Zn, as well as their oxides [14,15] and compounds [16,17,18,19,20,21,22]. These interests are related to their magnetic [23,24], optical [25], as well as conductive and semi-conductive [20,21,26] properties, which in turn translate to possible applications in catalytic, photocatalytic and photoelectrocatalytic (PEC) processes [14,27,28,29], as electrodes in photovoltaic cells and lithium-ion batteries [30,31,32], optoelectronic and plasmonic devices [15,25,33], gas sensors [34] or drug delivery [16,35].

For this reason, spectroscopic testing and characterization of these materials in the SXR spectral range near their L absorption edges, employing compact systems operating in the laboratory environment, has advantages, both in terms of the speed of the analysis, flexibility, and availability, without the immediate need to apply for the measurement time on the synchrotron as well as in utilizing the unique features of these edges. The testing of materials at the L edge is not intended to replace the information available at the K edge, but it should be a complimentary supplement [3,28]. The structure of the L edge is much richer than that of K [36,37,38] and thus, a much wider range of information can be obtained. Additionally, by combining the NEXAFS technique near the L edge with simulations, e.g., *fdmnes* or *FEFF* [39], we get a great research tool [36,38] extending its capabilities even further [40,41,42,43,44].

Moreover, the Xe/He laser-plasma source, employed in this work, demonstrates the suitability for NEXAFS measurements far above the “water window” energy range in a very compact and potentially widely available system. We show NEXAFS data obtained with this compact source for *3d* elements with Z = 26–30 (4th-period transition metals) deposited on top of a 100 nm thick SiN membrane, investigated in the transmission mode, according to the setup described previously in [45], but with much higher photon energy.

## 2. Experimental Setup

The experimental setup is depicted in Figure 1a and the photograph of the compact NEXAFS setup for the investigation of the transition metals is depicted in Figure 1b. The SXR radiation, used in this experiment for the acquisition of the NEXAFS spectra of the elements with Z = 26–30, is emitted from the laser-created plasma. The radiation with photon energy in the range of 220 to 1500 eV is delivered by a laser-plasma X-ray source (LPXS) based on a double stream Xe/He gas puff target. Such a laser-plasma, formed as a result of interactions of laser pulses with Xe gas, emits broad wavelength range radiation, from the soft X-rays to infrared. In this experiment, the application of an SXR emission from the xenon target was limited to the energy range of ~200–1500 eV (wavelength of 0.8 nm to 5.6 nm), as can be seen in Figure 2. The details about the SXR source were previously published in [45]. Herein, we are presenting only the necessary details about the source, required from the point of view of this work.

The SXR radiation is generated through the interaction of a compact Nd:YAG laser (NL 303 HT, EKSPLA, Vilnius, Lithuania, pulse energy of E_L_ = 0.6 J and 3 ns time duration at a repetition rate of 10-Hz) with a target. The radiation was focused by a 12.7 mm in diameter, 25 mm focal length best-form lens to a spot with ~100 µm diameter, reaching a power density in the focal point ~2.5 × 10^12^ W/cm^2^. The target was produced in a form of double stream Xe/He gas puff (cloud), where Xe and He backing pressures were optimized for maximum photon yield and were equal to 10 bar for each gas. The details about the geometry, timing [46], and synchronization can be found in [45] and are presented schematically in Figure 1a. The laser-target interaction conditions were chosen in such a way as to achieve the maximum photon yield in the SXR spectral region from 680–1100 eV, as can be seen in Figure 2. The yield was measured using the SXR spectrometer (MUT, Warsaw, Poland). The spectrum was obtained by integrating 40 X-ray pulses by the camera chip cooled down to −20 °C. A small circular aperture with ~5 mm in diameter, located 50 mm from the plasma was used to limit the reabsorption of the produced SXR radiation in the residual gas. It was achieved by separating the region of relatively high pressure inside the source chamber (P = 10^−2^ mbar during the source operation and P = 10^−4^ mbar otherwise) from the pressure inside the SXR spectrometer (typically 10^−5^–10^−6^ mbar). The gas puff target source provides the SXR energy used for NEXFAS studies with spectral stability of ~5% for the optimized source. Previous measurements resulted in estimation of the photon yield in the photon energy range from 440 eV to 1550 eV (selected by using a 3 µm thick Be filter) to be equal to 2.3 × 10^9^ photons/pulse/µm^2^ of the source surface emitted into a 4π solid angle. The SXR source size was 43.8 × 254.9 µm^2^ and an SXR pulse duration ~1.3 ns. For this experiment no filter has been used to increase the photon number, preserving, however, the spectral character of the SXR emission for parallel acquisition of the absorption spectrum. 

The emission spectra were recorded using a homemade SXR spectrometer, equipped with a grazing-incidence flat-field diffraction grating (Hitachi High Technologies America, Inc., Schaumburg, IL, USA). The spectrometer configuration was previously reported in [47]. The spectrometer is also equipped with a 100 µm wide entrance slit, located ~715 mm from the plasma source. For detection and storage of the SXR spectrum a GE 20482048 (greateyes, Berlin, Germany) CCD camera (2 k × 2 k chip, each pixel is 13 × 13 µm^2^ in size), as depicted in Figure 1a and Figure 3, was used. 

The SXR radiation from Xe/He plasma illuminates the thin sample located in a small load-lock downstream the beamline, as can be seen in Figure 3, which depicts the optical arrangement used to obtain simultaneously the sample and the reference beams. The sample holder allows illumination of the sample by the SXR light from the source. The SXR light is then transmitted through the sample forming a sample beam. Due to its design, the holder allows also some portion of the SXR wavefront, called here a reference beam, to directly enter, undisturbed, the spectrometer’s entrance slit. Thus, using this approach, contrary to other compact systems, i.e., [48], simultaneous acquisition of the two spectra has been achieved. We called this approach a dual-channel SXR spectrometer. Such a solution allows for the spectral acquisition and NEXAFS measurements to remain unaffected by the mechanical instabilities or the energy fluctuations of the source. These in turn can lead to unpredicted spectral shifts as well as serious difficulties in calculating the optical density in the presence of spatially or spectrally uncorrelated sample and reference signals.

The samples were prepared by the vacuum deposition method from metal pellets (ONYXMET, Olsztyn Poland) using e-beam evaporation (Syrus III 1100, Bühler Leybold Optics, Alzenau, Germany). The base pressure of the system was 3.5 × 10^−6^ mbar. The zinc sample was also prepared by e-beam evaporation; however, a different system was used (PVD 75, Kurt J. Lesker, St. Leonards-on-Sea, UK). The base pressure of the system was 2.7 × 10^−6^ mbar. The thicknesses of all layers were measured with quartz microbalance. Commercially available SiN membranes (Silson Ltd., Southam, UK), with a thickness of 100 nm and window size of 1.5 × 1.5 mm^2^ in a 5 × 5 mm^2^ frame were used as a target for deposition of various transition metals from period 4 of the periodic table of elements with thicknesses ranging from ~100 nm to ~220 nm. For this work, we selected iron (Fe), Z = 26, cobalt (Co), Z = 27, nickel (Ni), Z = 28, copper (Cu), Z = 29, and zinc (Zn), Z = 30. These metals, due to their electronic configurations, exhibit L absorption edges in the spectral region of interest from 680 eV to 1100 eV [49]. The characteristic parameters of the samples are listed in Tables 1 and 2. The supporting material -SiN has K and L absorption edges far out of the region of interest of this work, namely: Si-K (1839 eV), Si-L_1_ (149.7 eV), Si-L_2_ (99.8 eV), Si-L_3_ (99.2 eV), N-K (409.9 eV), N-L_1_ (37.3 eV), N-L_2_ (17.5 eV).

A typical procedure for the SXR spectrometer calibration was performed using the Ar:N_2_:O_2_ (1:1:1 by volume) gas mixture and SF_6_ gas injected into the laser-target interaction region. The positions of some most prominent emission lines spanning the spectral region of interest were found, namely, the fluorine line at λ = 1.6807 nm from F^8+^ ions, oxygen line λ = 2.1602 nm from O^6+^ ions, two SXR nitrogen lines: λ = 2.489 nm from N^6+^ (hydrogen-like) ions and λ = 2.878 nm from N^5+^ (helium-like) ions, and argon lines λ = 4.873 and λ = 4.918 nm from Ar^8+^ (neon-like) ions were used to construct the calibration curve. Additionally, the resolving power of the SXR spectrometer, in the spectral region of interest for this work, at the photon energy of 738 eV (isolated F line at λ = 1.6807 nm from F^8+^ ions with a measured spectral width of 3.7 eV) was equal to E/ΔE ~200.

## 3. Experimental Results

The reference spectrum emitted from Xe/He plasma for Xe and He gas backing pressures of 10 bar is presented in Figure 2. The blue region indicates the well-known “water window” spectral range that is typically used for SXR microscopy applications [50,51,52], but also can be used for NEXAFS studies of organic materials [10,53] (C-K and O-K edges are indicated at 284 and 540 eV respectively, N-K edge is in the middle of that range at ~410 eV). In this case, however, we are more interested in the higher energy range of the Xe emission. The gray region in Figure 2 indicates the main energy range of interest, 680–1100 eV, that is used in this work for L-edge NEXAFS of transition metals. The spectral range of our interest is dominated by one, most intense, high energy band from 850 eV to 950 eV, with the peak at ~900 eV, [45]. This emission originates from 3p^5^3d^10^4s, or 3p^6^3d^9^4f down to 3p^6^3d^10^ level in XeXXVII ions [54].

### 3.1. Thickness Influence on NEXAFS Measurements

The relation between the optical density spectrum *OD*(*E*) and the attenuation coefficient spectrum *Att*(*E*) is presented as Equation (1). The spectral attenuation coefficient was obtained from the optical density spectrum, taking into account the thickness of sample *d*:(1)$$Att(E)=\frac{OD(E)}{d}=-\frac{1}{d}\ln\left[\frac{S_{sam}(E)}{S_{ref}(E)}\right]$$
where *S_sam_*(*E*) is the sample spectrum and *S_ref_*(*E*) is the reference spectrum, as was explained using Figure 3. The attenuation spectrum and the optical density spectrum were achieved by slightly smoothing the raw data, obtained using Equation (1), using the Golay–Savitzky filter (order = 3 and frame length = 11, 3 × 11 filter) in a typical post-processing approach, described in [7]. A typical optical density spectrum (a) and attenuation coefficient spectrum (b) is presented in Figure 4 for various thicknesses of the sample. 40 SXR pulses were used to acquire a one CCD image with an exposure time of 5 s. A total of 16 images were added together to improve the signal-to-noise ratio by a factor of 4, except for a case of Cu L_2,3_ edges, where 25 images were integrated. Two regions, 200 pixels in width, in the CCD image, corresponding to the sample spectrum region and reference spectrum region were further integrated. This allowed us to extract the sample spectrum *S_sam_*(*E*) and the reference spectrum *S_ref_*(*E*) and use them to compute *OD*(*E*) and *Att*(*E*).

To demonstrate the spectral changes in the *OD*(*E*) and *Att*(*E*), a Ni sample was chosen with Ni layers of thicknesses ranging from 91 nm to 234 nm, as depicted in Figure 4. The sample thickness was measured using an atomic force microscope (AFM) (Veeco NanoScope IV MultiMode AFM, Veeco Metrology, Plainview, NY, USA), based on 10 cross-section measurements. The representative sample morphology is depicted in Figure 5 for the 138 nm thick Ni sample. The AFM scan was performed with a 20 × 20 µm^2^ scan size. As can be noted, changing the sample thickness preserves all the main features of the attenuation spectrum, as well as the positions of the L_3_ and L_2_ edges. 

### 3.2. NEXAFS L-Edge Spectra of Z = 26–30 Transition Metals

As a demonstration in the course of this work, the transition metals from period 4 of the periodic table of elements were studied with atomic number Z between 26 and 30. The list of elements studied as well as the sample thicknesses are listed in Table 1. There are several commonly accepted methods for determining the position of the absorption edge, depending on the quality of the data obtained. One of them is the determination of the edge position relative to the 50% slope, and this is what was used in this study, similarly to, e.g., in [38,55].

In the case of iron (Fe, Z = 26) all three L absorption edges were identified, as depicted in Figure 6a,b. We have found that the energies of the experimental L edges E_e_ [1, 2, 3] = [844.2, 719.0, 706.3] eV are slightly different from the theoretical energies of the L edges in bulk material, [49], E_t_ [1, 2, 3] = [844.6, 719.9, 706.8] eV, with the energy difference of ΔE = E_t_ − E_e_, equal to ΔE [1, 2, 3] = [0.4, 0.9, 0.5] eV. 

In the case of cobalt (Co, Z = 27), we could not identify the L_1_ absorption edge, due to too low absorption in the absorbing Co layer. For this reason, only the energies of the experimental L_2,3_ edges were found, Figure 6c and are equal to E_e_ [2, 3] = [794.0, 778.0] eV. The energies of the theoretical L edges, [49], are equal to E_t_ [2, 3] = [793.3, 778.1] eV, with the energy difference of ΔE [2, 3] = [−0.7, 0.1] eV. 

A similar case can be observed in nickel (Ni, Z = 28), where the L_1_ edge was also not visible, below the sensitivity threshold of our setup. In the case of nickel, the energies of the experimental L_2,3_ edges are presented in Figure 6d and were measured to be E_e_ [2, 3] = [869.5, 852.4] eV. The energies of the theoretical L_2,3_ edges, [49], are equal to E_t_ [2, 3] = [870.0, 852.7] eV, with the energy difference of ΔE [2, 3] = [0.5, 0.3] eV. 

For copper (Cu, Z = 29) all L_1_ to L_3_ edges were visible, as depicted in Figure 6e,f. The energies of the experimental L edges were measured to be E_e_ [1, 2, 3] = [1098.0, 949.7, 938.2] eV and the energies of the theoretical L edge, [49], were equal to E_t_ [1, 2, 3] = [1096.7, 952.3, 932.7] eV, with the energy difference of ΔE [1, 2, 3] = [−1.3, 2.6, −5.5] eV. 

Finally, absorption edges of zinc (Zn, Z = 30) were also identified with a similar data trend as in the case of cobalt and nickel. The energies of the experimental L_2,3_ edges, presented in Figure 6g, were measured to be E_e_ [2, 3] = [1048.0, 1021.0] eV and the energies of the theoretical L edges, [49], are equal to E_t_ [2, 3] = [1044.9, 1021.8] eV, with the energy difference of ΔE [2, 3] = [−3.1, 0.8] eV. 

Possible differences in the energies arise due to the limited spectral resolution of our spectrometer, and hence an additional equipment broadening might also influence the energies slightly (of the order of ~0.1 eV to ~1 eV at a photon energy of ~1 keV, corresponding to 0.01% to 0.1% energy shift for most of the samples). Our sample is not ideal bulk material, but a thin layer deposited on the top of the supporting SiN membrane [23] could be another source of deviation. Importantly, we compared our experimental results so far with only one database [49]. In the literature, however, there are other sources of data reporting the L edges at slightly different energies, e.g., Kaye & Labe [56] or NIST database [57]. In the case of copper, for which we have the largest ΔE, the L_1_ edge is at 1103.12 eV, L_2_ at 959.58 eV, and L_3_ at 939.85 eV [58], resulting in better matching to our experimental data regarding the L_3_ edge (ΔE = 1.7 eV, or 0.15%), however, increasing the discrepancy for L_2_ edge up to ΔE = 9.9 eV (0.89%).

Additionally, the Laser-Induced Breakdown Spectroscopy (LIBS) was used to detect the elemental composition of the samples. For that purpose witness samples were used because samples on the SiN membrane were too fragile and easily destroyed during the measurement. The witness samples were obtained by depositing the metal layers on top of 10 × 10 mm^2^ Si wafers in the same deposition process (same conditions) as the NEXAFS samples (on SiN membrane support) 5 cm away from each other and at an equal distance from the electron beam spot. In LIBS measurements a 20 Hz 1064 nm laser source (Quantel, Bozeman, MT, USA, BigSky Brio model) with 50 mJ/4 ns pulses was used. The 4 mm in diameter laser beam was directed through a 150 mm focusing lens onto the investigated samples. The lens-to-sample distance was fixed at 145 mm to avoid an accidental plasma formation in the air (laser spark). The energy of a single laser pulse was sufficient to acquire a useful spectrum. Plasma radiation was registered by ESA 4000 Echelle spectrometer equipped with an ICCD camera with Kodak KAF 1001 detector (spectral resolution of the system was about λ/Δλ~20,000). This experimental system has been used before in different experiments with solid or gaseous targets [59,60,61,62]. 

To capture useful line radiation, data collection lasted 500 ns and began 500 ns after the laser pulse, allowing the initial continuum emission from the hot plasma to weaken significantly. LIBS measurements showed only the main sample material, due to the high purity of metals deposited on the SiN membrane, without any detectable impurities, and did not provide quantitative results. For this reason, a more quantitative analysis was performed by Energy Dispersive Spectroscopy (EDS). The EDS analysis using a Quanta 250 FEG SEM (FEI, Hillsboro, OR, USA) was also performed on witness samples in three different locations. The acquisition parameters of the EDS are an accelerating voltage of 30 kV, spot 4.5 corresponding to the spot diameter on the sample equal approximately to 2 µm.

Table 2 shows the results of EDS analysis on all samples. All samples investigated using the EDS show a strong signal from the interaction volume originating from the supporting Si membrane, as well as a weaker signal from the investigated target material ranging from 7 to ~20% (by weight). Moreover, no additional elements (admixtures or infusions) were detected by the LIBS and EDS. 

In the case of copper (~163 nm thick, comparable to cobalt sample) all L edges were detected at the amount of ~11% of copper in the investigated by the EDS volume using our compact NEXAFS setup. Copper L_1_ edge occurs also at the highest recorded in this experiment energy at ~1.1 keV, demonstrating the applicability of gas-based laser plasma source to such measurements above 1 keV photon energy. Copper is also second in thickness among the samples, second only to zinc sample, providing good absorption for SXR radiation. Even better results were obtained for the iron sample (~142 nm thick), in which only ~7.5% of iron was measured in the EDS interaction volume while providing data for all three L edges. Although there was more cobalt in the cobalt sample (~160 nm thick), on average 14.2% by weight, the L_1_ edge was not detected, probably due to the insufficient signal-to-noise ratio of the absorption spectral data obtained using our system, caused mainly by limited spectral resolution of the spectrometer. A similar case is for the nickel at ~9.2% for 140 nm thick sample and zinc at almost 19.5% for quite thick 300 nm sample for which only L_2,3_ edges were found. 

The L_2,3_ edges for all samples (single layers and multilayers) were detected; however, some L_1_ edges (for cobalt, nickel, and zinc) were not visible. We believe it is probably due to insufficient signal-to-noise ratio of the absorption spectral data, caused by the fact that the L_2,3_ edges are typically 10–100 times more pronounced (higher value of the ratio of the OD(E) below and above the absorption edge) than the L_1_ edges [3].

### 3.3. NEXAFS Spectrum of a Multilayer Sample

To demonstrate the possibility to study multi-component materials a multilayer sample composed of Fe, Ni, and Co was also studied. To prepare the sample Co, Ni, and Fe were subsequently deposited on top of a 100 nm thick SiN membrane. Theoretically, each layer should have a thickness of 70 nm; however, the total thickness of the three layers amounted to 216 nm, slightly higher than expected, with a standard deviation of 3.5 nm. The small difference in the thickness was caused by the deposition process. The NEXAFS spectrum of such a multilayer sample is presented in Figure 7.

In the case of iron, all three L edges are visible, as in the case of a single layer sample. The energies of the L edges E_e_ [1, 2, 3] obtained in the experiment were [844.3, 719.0, 702.8] eV, and differed from the theoretical energies by [0.3, 0.9, 4.0] eV, while the difference between multilayer sample energies E_eM_ and single layer energies E_eL_, ΔE_2_ = E_eM_ − E_eL_, was equal to ΔE_2_ = [0.1, 0.0, −3.5] eV. The pre-edge feature near the Fe-L_3_ edge was identified and it was Xe-M_4_ edge at 689 eV. The origin of that feature is the residual amount of Xe gas along the SXR beam path from the source to the spectrometer, while the gas itself originates from the Xe/He gas puff target.

In the case of cobalt, as for a single layer sample, only L_2,3_ edges were visible. The energies of the experimental L_2,3_ edges were equal to E_e_ [2, 3] = [793.5, 777.7] eV, and differ from the theoretical energies by [0.2, 0.4] eV, while the energy difference ΔE_2_ = [−0.5, −0.3] eV.

Finally, in the case of nickel, also only L_2,3_ edges were visible, with the energies of the experimental L_2,3_ edges measured to be E_e_ [2, 3] = [869.8, 852.6] eV. The difference from the theoretical energies were [0.2, 0.1] eV, while the energy difference ΔE_2_ = [0.3, 0.2] eV.

The ΔE_2_ values are generally small (<0.5 eV), except for the L_3_ absorption edge of iron, where it exceeds 3 eV. Table 3 summarizes the data for the Co-Ni-Fe sample.

## 4. Conclusions

In conclusion, we have shown the possibility to study the L absorption edges of transition metals from the 4th period of the periodic table of elements with Z numbers ranging from 26 to 30, including the most important elements exhibiting magnetic properties. The proof experiment employed the SXR emission from a laser-plasma source covering a spectral region near 1-keV. The laser plasma source was based on a double stream Xe/He gas puff target. We have shown that using a dual channel NEXAFS measurement system, recording independently the sample and reference spectra, it was possible to acquire good quality spectra from single and multilayered samples in the transmission mode reaching Cu-L_1_ edge around 1.1 keV. Compact NEXAFS was demonstrated before reaching photon energies above 1 keV, e.g., [7]; however, it was performed with solid-state targets, e.g., copper, producing debris from the laser ablation process. The debris prohibits measurements of delicate samples and, with long exposures, may cover the measured samples with an additional metal layer. This layer is originating from target debris, affecting in turn, the quality of the measured spectra. Additionally, the spectrometer used for obtaining NEXAFS data employed two separate (and not identical) dispersion elements—complicated, narrow energy range off-axis zone plates, also affecting the data quality. The NEXAFS data we have shown were not demonstrated so far using a compact laser-plasma system based on a gaseous target, and the dual-channel spectrometer setup we have developed was also employed in this energy range for the first time. The presented results demonstrated that all essential features of the collected absorption spectra remain the same for various sample thicknesses (the nickel data) and the system is stable providing proper calibration in the spectral domain (stable positions of the absorption peaks). The demonstrated absorption spectra near the L_1–3_ absorption edges corresponded to transition metals, such as iron, cobalt, nickel, copper, and zinc. The L_2,3_ edges were detected for all samples (single and multilayers). However, some L_1_ edges were not detected in the experiment, in particular the L_1_ edge of cobalt, nickel, and zinc, probably due to the insufficient signal-to-noise ratio of the absorption spectral data. The reason for this seems to be the fact that L_2,3_ edges are typically significantly stronger than the L_1_ edges [3]. This work proves the possibility to apply a compact gas-based laser plasma source to the NEXAFS technique in a much broader than usual spectral range. As a consequence, the results demonstrate the applicability of the presented technique to spectroscopic studies not only organic materials but also metals, including, important from the industrial point of view, magnetic materials (Co, Ni, Fe), typically studied using synchrotron sources.

## Figures and Tables

**Figure 1 materials-14-07337-f001:**
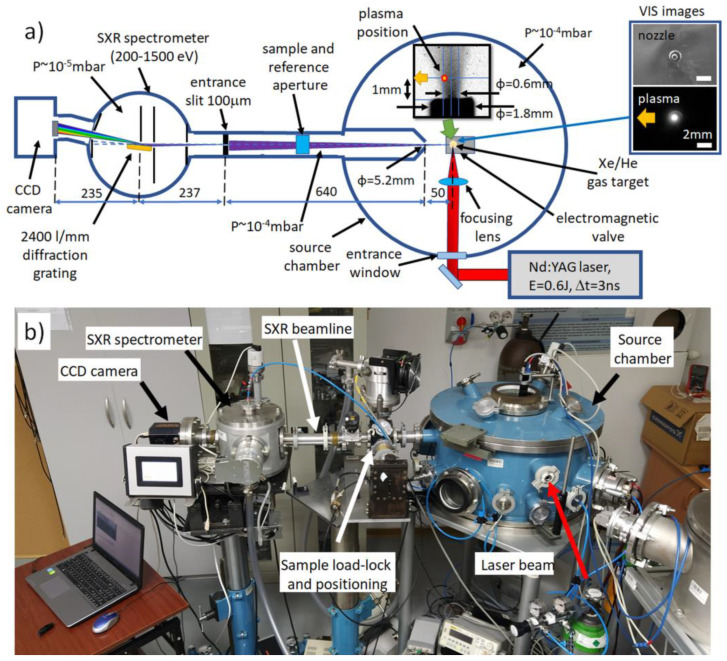
Schematic representation (**a**), modified from [45], and the photograph (**b**) of the experimental setup for transition metals NEXAFS measurements using near 1 keV from laser-produced Xe plasma, indicating major components. Orange arrows indicate the radiation propagation direction to the SXR spectrometer.

**Figure 2 materials-14-07337-f002:**
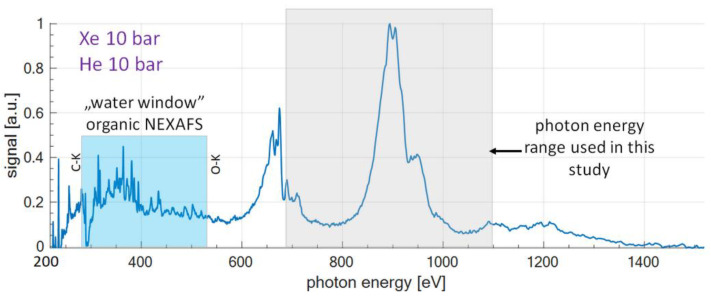
SXR reference spectrum in the photon energy range of 200–1500 eV, obtained for Xe and He gas backing pressures of 10 bar. The Blue region indicated well known “water window” spectral range that can be used for NEXAFS of organic materials (C and O-K edges are indicated). Gray region indicates 680–1100 eV region used for L-edge NEXAFS of transition metals, reported in this study.

**Figure 3 materials-14-07337-f003:**
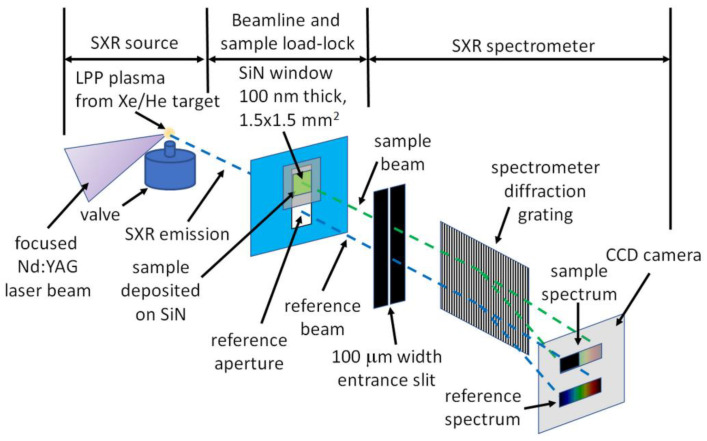
Scheme of the dual-channel SXR NEXAFS approach, indicating sample and reference SXR beams forming two independent spectra on the CCD camera. The spectra are used for the calculation of the OD spectrum.

**Figure 4 materials-14-07337-f004:**
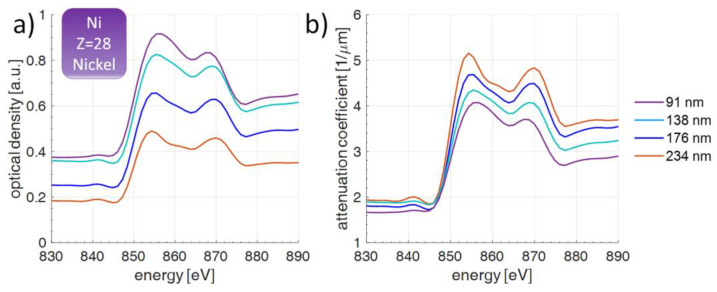
Optical density (**a**) and attenuation coefficient (**b**) spectra in the photon energy range from 830 eV to 880 eV for thin layers of Ni of various thicknesses, deposited on top of 100 nm thick SiN membrane. The presented spectra are obtained using data filtering by the Golay–Savitzky 3 × 11 filter.

**Figure 5 materials-14-07337-f005:**
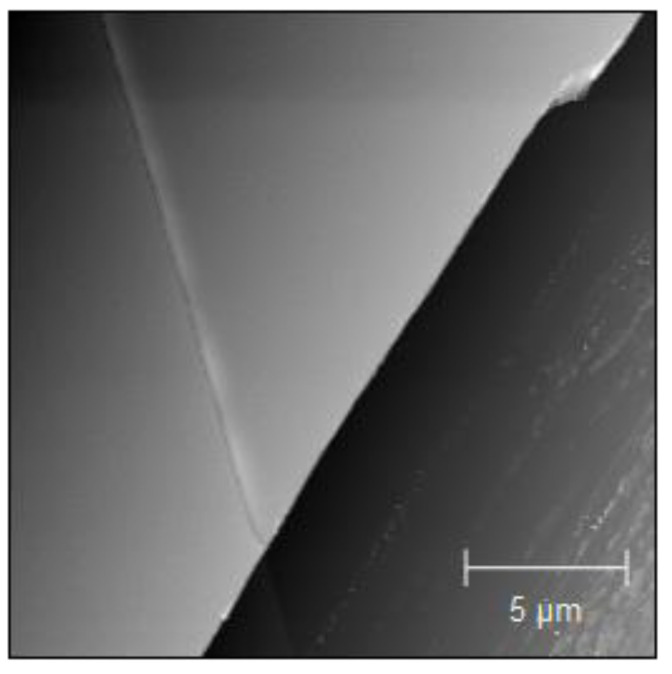
AFM scan of a Ni sample 138 nm thick, scratched for thickness estimation. On the left, a Ni layer is visible, and on the right—a Si wafer. A 20 × 20 µm^2^ image with a resolution of 256 × 256 pixels is showing typical sample morphology.

**Figure 6 materials-14-07337-f006:**
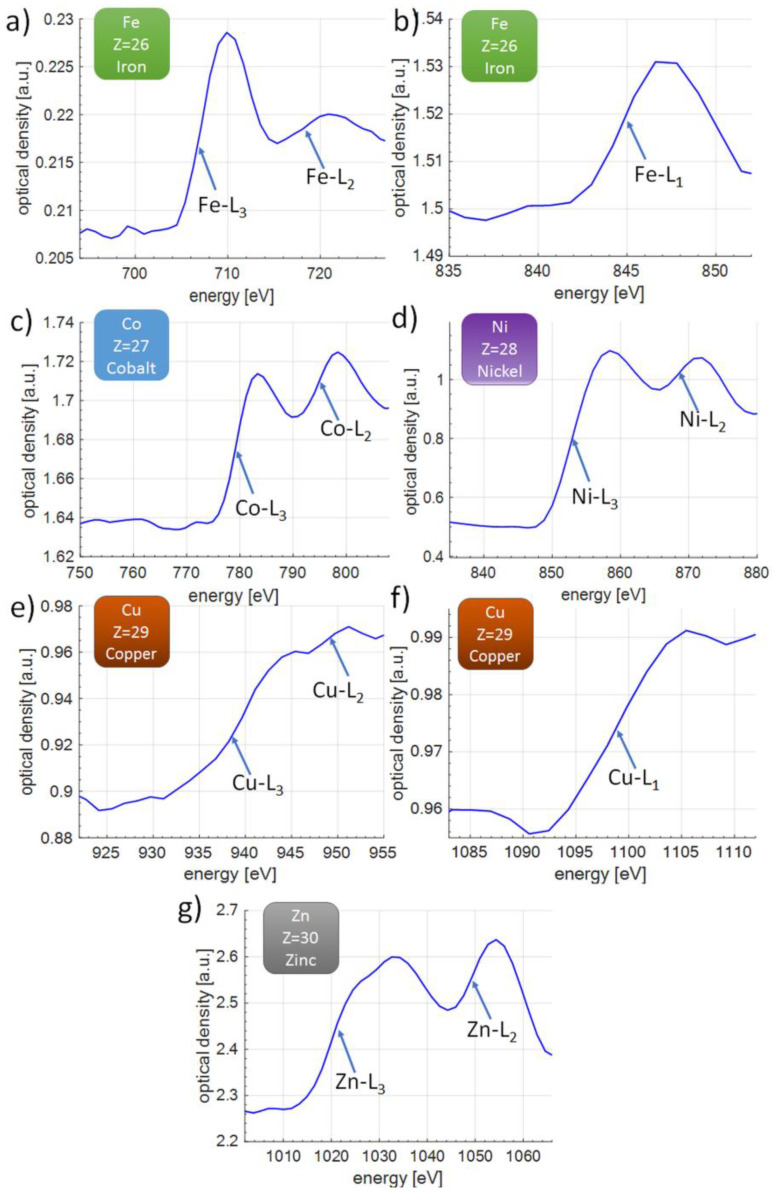
Overview of L-edge NEXAFS spectra for transition metals from Z = 26 to 30, in the photon energy range from 680 eV to 1.1 keV: iron (**a**,**b**), cobalt (**c**), nickel (**d**), copper (**e**,**f**), and zinc (**g**). Sample thicknesses are listed in Table 1.

**Figure 7 materials-14-07337-f007:**
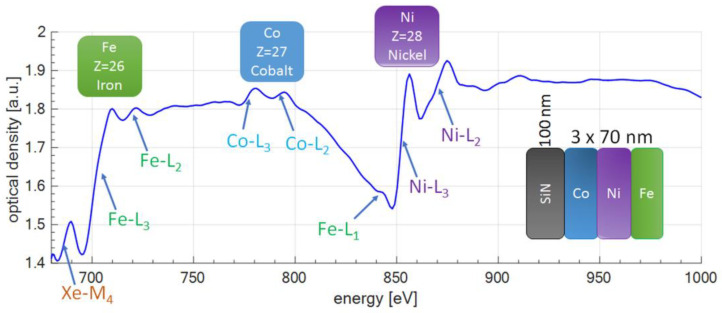
The wide-band spectral optical density of a Fe-Co-Ni multilayer sample (each layer ~70 nm in thickness), total sample thickness ~216 nm. The figure shows identifiable L absorption edges of all three sample constituents.

**Table 1 materials-14-07337-t001:** List of transition metals deposited on top of 100 nm thick SiN membranes as single layers, as well as the results of L_1_–L_3_ edge NEXAFS energies: theoretical (E_t_), based on [49], and experimental values (E_e_). Some of the edges were not detected in this experiment. Experimentally obtained edge energies are slightly shifted compared to theoretical energies for bulk materials, as described by the ΔE parameter, where ΔE = E_t_ − E_e_.

Element /Sample	Z Number	Thickness [nm]	Energy E_t_—Theory E_e_—Experiment ΔE—Difference	Absorption Edges [eV]		
				L_1_	L_2_	L_3_
Fe	26	142.3 ± 2.1	E_t_	844.6	719.9	706.8
			E_e_	844.2	719.0	706.3
			ΔE	0.4	0.9	0.5
Co	27	158.2 ± 3.0	E_t_	925.1	793.3	778.1
			E_e_	-	794.0	778.0
			ΔE	-	−0.7	0.1
Ni	28	138.0 ± 2.1	E_t_	1008.6	870.0	852.7
			E_e_	-	869.5	852.4
			ΔE	-	0.5	0.3
Cu	29	163.1 ± 0.9	E_t_	1096.7	952.3	932.7
			E_e_	1098.0	949.7	938.2
			ΔE	−1.3	2.6	−5.5
Cu *	29	163.1 ± 0.9	E_t_	1103.1	959.6	939.9
			E_e_	1098.0	949.7	938.2
			ΔE	5.1	9.9	1.7
Zn	30	309.2 ± 9.0	E_t_	1196.2	1044.9	1021.8
			E_e_	-	1048.0	1021.0
			ΔE	-	−3.1	0.8

* comparing to NIST database [57,58].

**Table 2 materials-14-07337-t002:** EDS analysis results.

		EDS Composition			
Sample	Thickness [nm]	wg%	at%	wg%	at%
		Si		Fe	
Fe	142.3 ± 2.1	92.52 ± 0.33	96.09 ± 0.18	7.48 ± 0.33	3.91 ± 0.18
		Si		Co	
Co	158.2 ± 3.0	85.83 ± 0.06	92.71 ± 0.03	14.17 ± 0.06	7.29 ± 0.03
		Si		Ni	
Ni	90.8 ± 0.4	92.76 ± 0.09	96.40 ± 0.05	7.24 ± 0.09	3.60 ± 0.05
	138.0 ± 2.1	90.80 ± 0.18	95.37 ± 0.10	9.20 ± 0.18	4.63 ± 0.10
	234.0 ± 1.4	81.60 ± 0.08	90.26 ± 0.05	18.40 ± 0.08	9.74 ± 0.05
		Si		Cu	
Cu	163.1 ± 0.9	88.97 ± 0.30	94.81 ± 0.14	11.03 ± 0.30	5.19 ± 0.14
		Si		Zn	
Zn	309.2 ± 9.0	80.53 ± 0.31	90.59 ± 0.17	19.47 ± 0.31	9.41 ± 0.17

**Table 3 materials-14-07337-t003:** Co-Ni-Fe multilayer sample. The results of L_1–3_ edge NEXAFS experimentally obtained energies for multilayer and single layers of constituent materials. Experimentally obtained edge energies are slightly shifted comparing single metal layer and multilayer, where ΔE = E_eM_ − E_eL_.

Multilayer	Z Number	Thickness [nm]	Experimental Edges (Multilayer) E_eM_ [eV]		
			L_1_	L_2_	L_3_
Co-Ni-Fe	26–28	215.7 ± 3.5 (3× ~70 nm)	Fe: 844.3	719.0	702.8
			Co: -	793.5	777.7
			Ni: -	869.8	852.6
**Constituents**			**Experimental edges (layer) E_eL_ [eV]**		
Fe	26	142.3 ± 1.9	844.2	719.0	706.3
Co	27	158.2 ± 3.0	-	794.0	778.0
Ni	28	138.0 ± 2.1	-	869.5	852.4
**Energy difference**			**ΔE_2_ = E_eM_ − E_eL_ [eV]**		
Fe	26	142.3 ± 1.9	0.1	0	−3.5
Co	27	158.2 ± 3.0	-	−0.5	−0.3
Ni	28	138.0 ± 2.1	-	0.3	0.2

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
