# Peer review of "Demonstration of Near Edge X-ray Absorption Fine Structure Spectroscopy of Transition Metals Using Xe/He Double Stream Gas Puff Target Soft X-ray Source"

_materials, 2021, doi:10.3390/ma14237337_

Round 1

Reviewer 1 Report

This paper is a study of the Demonstration of near-edge X-ray absorption fine structure spectroscopy of transition metals using Xe/He double stream gas puff target soft X-ray source.

The paper is well written, the English are good. 

1. In the “experimental results” the part of measuring the thickness of the films should be compared with X-ray reflectometry measurements.
2. In the “experimental results” morphology of the film surface would be useful (SEM or AFM).

I recommend minor revision.

Reviewer 2 Report

Reviewer report

Paper: Materials- 1462698

Demonstration of near edge X-ray absorption fine structure spectroscopy of transition metals using Xe/He double stream gas puff target soft X-ray source” by Tomasz Fok, Przemysław Wachulak, Łukasz Węgrzyński, Andrzej Bartnik , Michał Nowak, Piotr Nyga , Jerzy Kostecki , Barbara Nasiłowska, Wojciech Skrzeczanowski, Rafał Pietruszka, Karol Janulewicz, Henryk Fiedorowicz .

This work deals with experimental set-up and results of near edge X-ray absorption fine structure spectroscopy (NEXAFS) applied to 3d transition metals, from Z = 26 to 30, using Xe/He double stream gas puff target as soft X-ray source.

 This NEXAFS technique is of great interest and many works have been conducted over the years in order to reveal the physical origin of numerous properties of materials. Much research have been conducted in the last years in order to develop laboratory/home-made NEXAFS systems.

Although not entirely new, the authors present here good experimental work, and on the whole, the manuscript is well-organized. The text is well-written and Figures are in general clear and completed.

I recommend the publication of this paper, but before some questions and comments arise:

1.- Abstract: I recommend remove the phrase about the influence of the thickness as not much is shown in the paper.

2.- Experimental results: page 9, line 305. Can authors explain in more detail the results coming from the Cu case?

3.- Experimental results: page 10, line 276. Can authors show and explain the LIBS results and a comparison with those of EDS?

4.- Experimental results: I would like authors to compare their experimental results with those obtained by others researchers using the same technique, highlighting their novelty.

5.- Conclusions: Must be rewritten highlighting the novelty of the work, removing the phrase on the influence of the thickness since almost no results on it have been shown.

Reviewer 3 Report

The paper entitled “Demonstration of near edge X-ray absorption fine structure spectroscopy of transition metals using Xe/He double stream gas puff target soft X-ray source” by T. Fok et al. provides a comprehensive overview about the possibility to study the L absorption edges in transition metals.

In my opinion the paper reports some valuble insights which worth publication. Minor Comments to futher improve this paper are listed below:

  1. Lines 23-24: Please, a re-phrase of this sentence is required.
  2. Lines 308-311: "Although there was more cobalt in the cobalt sample (~160 nm thick), on average 14.2 % by weight, the L1 edge was not visible. A similar case is for the nickel at ~9.2 % for 140 nm thick sample and zinc at almost 19.5 % for quite thick 300 nm sample for which only L2,3 edges were found. ". Why, in authors opinion, in cobalt  and nickel samples the L edges were not visible?  The limited spectral resolution of the spectrometer could lead to variation in the energy values associated to observed transition (compared to those found in databases), as corretly stated by authors, but not to the absence of specific edges. Please try to explain such aspect in the text in more detail since it could be helpull for the reader (also in terms of reproducibility of the experiment).
  3. Lines 318-319: What does "As  a  proof  of  principle" means for authors?. Please, a re-phrase of this sentence is strongly recommended.
  4. I would suggest a careful review of the English style by a native English speaker.

Reviewer 4 Report

Manuscript contains important and interesting results, but minor editing is desirable:

  1. Abbreviation "PEC" (page 2, line 59) should be explained.
  2. Possible reasons of difference of experimental and theoretical values of L edges, mentioned for the first time on page 7 (lines) 236-239, are considered on page 9 (lines 264-275). Therefore, it seems to be desirable to write "Possible reasons are considered below" or something similar after the sentence on lines 236-239. 
  3. It is desirable to insert the word "registered" or similar one before the word "absorption" on page 11, line 363.
